# FreeRet: MLLMs as Training-Free Retrievers

## Abstract

Multimodal large language models (MLLMs) are emerging as versatile foundations for mixed-modality retrieval. Yet, they often require heavy post-hoc training to convert them into contrastive encoders for retrieval. This work asks: *Can off-the-shelf MLLMs serve as powerful retrievers without additional training?* We present **FreeRet**, a plug-and-play framework that turns any MLLM into a two-stage retriever. FreeRet first derives semantically grounded embeddings directly from the model for fast candidate search, and then exploits its reasoning ability for precise reranking. The framework contributes three advances: bypassing lexical alignment layers to obtain semantically faithful embeddings, conditioning representation generation with explicit priors, and mitigating framing effect in reranking via neutral choice framing. On the MMEB and MMEB-V2 benchmarks spanning 46 datasets, FreeRet substantially outperforms models trained on millions of pairs. Beyond benchmarks, FreeRet is model-agnostic and scales seamlessly across MLLM families and sizes, preserves their generative abilities, supports arbitrary modality combinations, and unifies retrieval, reranking, and generation into end-to-end RAG within a single model. Our findings demonstrate that pretrained MLLMs, when carefully harnessed, can serve as strong retrieval engines without training, closing a critical gap in their role as generalists.

## 1 Introduction

Multimodal retrieval, the process of retrieving relevant items across multiple modalities, is a cornerstone of modern AI systems. It underlies applications ranging from web search (Mitra et al., 2017; Huang et al., 2020) and retrieval-augmented generation (RAG) (Lewis et al., 2020; Gao et al., 2023), to embodied agents (Singh et al., 2025) and personalized recommendation (Rajput et al., 2023). Conventional solutions rely on two stages: embedding-based candidate search followed by reranking for accuracy. CLIP-style dual encoders (Radford et al., 2021; Li et al., 2022) have long been the workhorse of this paradigm, but they exhibit fundamental limitations: they struggle with long queries, compositional semantics, and interleaved multimodal inputs. These shortcomings highlight the need for a more generalizable foundation.

Multimodal large language models (MLLMs) offer such a foundation. With powerful reasoning and flexible input handling, they promise to unify understanding across modalities. Yet most recent efforts adapt MLLMs to retrieval through heavy post-hoc fine-tuning (Fig. 1(a)) (Lin et al., 2024; Zhang et al., 2024b; Liu et al., 2025b). This paradigm, however, encounters two persistent obstacles. First, it demands massive paired data and expensive fine-tuning for every new backbone or modality configuration, hampering scalability. Second, its generalization remains fragile: without in-domain supervision, even large, carefully curated models often perform poorly on standard benchmarks.

*Can we instead harness MLLMs for retrieval **without** training, letting them act simultaneously as embedders and rerankers?* Early training-free attempts embed generated token states (Jiang et al., 2024a; 2023), but these representations are coarse and generally omit reranking, a critical stage for robust performance. Other approaches integrate learned rerankers, but they sacrifice efficiency and modularity by requiring additional supervision and model components.

To address this gap, we propose **FreeRet**, a plug-and-play framework that transforms any off-the-shelf MLLM into a competitive two-stage retriever (Fig. 1(b)). FreeRet first extracts embeddings for efficient candidate search, then prompts the same model to conduct fine-grained reranking. Crucially, it requires no parameter updates, auxiliary models, or external data. By fully exploiting both

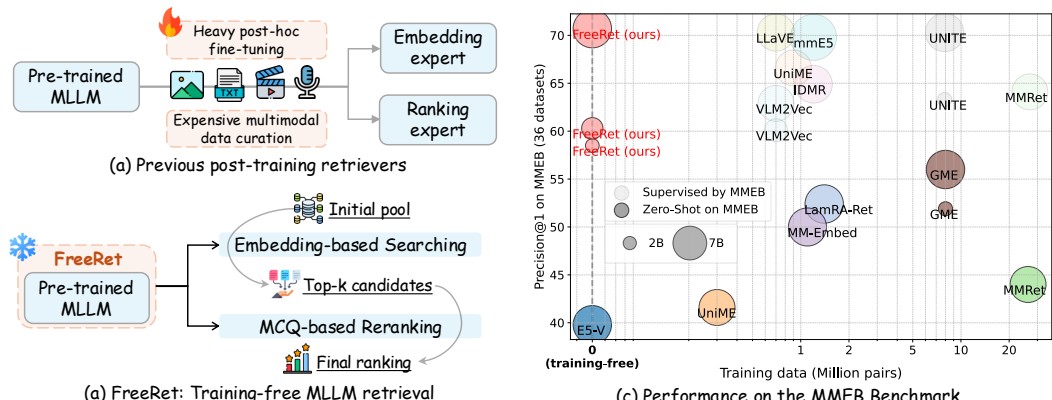

Figure 1: **Comparison between prior post-training retrievers and our FreeRet.** (a) Existing methods rely on extensive data curation and costly fine-tuning to construct *separate* embedding and reranking modules. (b) FreeRet directly employs MLLMs as *unified* embedders and rerankers without any extra training. (c) On the MMEB benchmark covering 36 datasets, FreeRet outperforms models trained on millions of pairs and matches the best methods supervised directly on MMEB.

the representational and reasoning capacities of MLLMs, FreeRet demonstrates that *training-free retrieval is not only feasible but also state-of-the-art competitive*.

Our framework rests on three key contributions: (1) We refine embedding quality by bypassing the final MLP before the LM head, which enforces surface-level lexical alignment at the expense of semantic structure. Removing it yields embeddings that better capture underlying meaning. (2) We stabilize the embedding space via controlled summarization prompts that inject semantic, denoising, and contextual priors. This improves semantic focus and task relevance. (3) We uncover an LLM *framing effect* in reranking: logically equivalent label formats (*e.g.* Yes/No vs. True/False) yield divergent accuracies due to pretraining biases (Zhao et al., 2021). We mitigate this by casting reranking as multiple-choice questions (MCQ), which elicit more neutral and reasoned judgments.

Evaluated on MMEB (Jiang et al., 2024b), a comprehensive suite of 36 datasets across four meta-tasks, FreeRet consistently delivers strong gains (Fig. 1(c)). Notably, FreeRet-2B outperforms GME-7B (Zhang et al., 2024b), trained on 8M multimodal pairs, and dramatically surpasses MMRet-7B (Zhou et al., 2024), trained on 26.2M pairs. Remarkably, FreeRet-7B competes head-to-head with methods explicitly supervised on MMEB, while on the video subset of MMEB-V2 (Meng et al., 2025), it exceeds even post-trained approaches by large margins. Together, these results highlight the strength of a purely training-free paradigm.

Beyond empirical gains, FreeRet delivers broader implications. It eliminates the costly adaptation barrier, allowing immediate deployment of any MLLM across scales and architectures (Tab. 1). It naturally supports arbitrary modality combinations when applied to omni-modal models like Qwen2.5-Omni (Xu et al., 2025a) (Fig. 6). By avoiding fine-tuning, FreeRet fully retains the instruction-following, conversational, and reasoning capacity of pretrained MLLMs. Moreover, it streamlines the RAG framework, unifying retrieval, reranking, and generation within a single model, and enhances long video understanding by retrieving relevant clips before deeper reasoning (Tab. 4). Additionally, FreeRet serves as a diagnostic lens, extending the evaluation of MLLMs beyond conventional QA settings to include retrieval-oriented tasks.

In summary, FreeRet establishes pretrained MLLMs as competitive, versatile, and training-free retrieval systems. It challenges the necessity of large-scale supervised adaptation and points toward a future where a single generalist model unifies retrieval with broader multimodal understanding.

## 2 RELATED WORK

**Multimodal Large Language Models (MLLMs).** The evolution of MLLMs reflects a steady progression from alignment to generation. CLIP (Radford et al., 2021) demonstrated the power of contrastive pretraining on web-scale image–text pairs, yielding highly transferable embeddings. BLIP (Li et al., 2022) advanced this framework by unifying contrastive and generative objectives.

Flamingo (Alayrac et al., 2022) introduced gated cross-attention for few-shot multimodal learning, while LLaVA (Liu et al., 2023; Li et al., 2024) pushed instruction tuning into visual dialogue to enhance interactive reasoning. More recent efforts, including GPT-4V (Achiam et al., 2023), Qwen-VL (Wang et al., 2024a; Bai et al., 2025), and InternVL (Chen et al., 2024; Zhu et al., 2025), scale these ideas toward general-purpose multimodality, addressing tasks from grounded reasoning to document-level understanding. Collectively, these advances trace a trajectory from multimodal alignment to complex reasoning, establishing the backbones for multimodal retrieval.

**Training-based Retrieval with MLLMs.** Early multimodal retrieval has largely relied on CLIP-style dual encoders (Radford et al., 2021; Zhai et al., 2023; Sun et al., 2023), yet this paradigm faces two core challenges. First, text encoders struggle with long or compositional inputs that demand fine-grained reasoning. Second, the unimodal design prevents robust handling of interleaved content, limiting applications such as composed image retrieval (Vo et al., 2019). To address these shortcomings, recent work repurposes MLLMs as retrieval backbones, treating MLLMs as universal encoders, and fine-tunes them with contrastive learning. Within this framework, researchers have explored a spectrum of techniques, ranging from hard negative mining (Gu et al., 2025a; Schneider et al., 2025; Lan et al., 2025; Xue et al., 2025) and modality-aware sampling (Lyu et al., 2025; Kong et al., 2025) to reinforcement learning (Zhao et al., 2025) and joint optimization with generative objectives (Ouali et al., 2025; Yu et al., 2025). Parallel lines of work focus on curating large-scale mixed-modality corpora (Zhang et al., 2024b; Zhou et al., 2024; Chen et al., 2025; Liu et al., 2025a), designing broad evaluation suites (Lin et al., 2024; Jiang et al., 2024b; Xiao et al., 2025), training rerankers (Xu et al., 2025b), and building end-to-end pipelines that combine embedding with reranking (Lin et al., 2024; Liu et al., 2025b; Cui et al., 2025). Despite these advances, training-based approaches retain fundamental bottlenecks. They demand massive multimodal datasets and expensive re-training whenever a new backbone or modality configuration is introduced. More importantly, models optimized for one benchmark (*e.g.* MMEB) often transfer poorly to others (*e.g.* MIEB), exposing weak generalization without benchmark-specific supervision.

**Training-free Retrieval with Auto-Regressive Models.** Most attempts at training-free retrieval with language models have mostly focused on the text-only setting. They extract embeddings from internal hidden states without further optimization. Early approaches such as PromptEOL (Jiang et al., 2023) design handcrafted prompts (*e.g.*, "The sentence means in one word:") so that the hidden state of the next token approximates a sentence-level representation. MetaEOL (Lei et al., 2024) and GenEOL (Thirukovalluru & Dhingra, 2024) improve robustness by combining multiple embeddings, while Zhang et al. (2024a) enhances expressivity using chain-of-thought prompting or external knowledge injection. Token-Prepend (Fu et al., 2024) and Echo-Embedding (Springer et al., 2024) mitigate the causal attention bias that suppresses early tokens, while MoEE (Li & Zhou, 2024) fuses routing weights with hidden states in MoE architectures to yield more expressive embeddings. In contrast, the multimodal setting remains underexplored (Jiang et al., 2024a; Ju & Lee, 2025). They often produce coarse representations that capture only partial cross-modal alignment, and they lack integration with reranking mechanisms which is crucial to accurate retrieval. As a result, their performance lags far behind approaches that permit task-specific training.

## 3 FREERET

### 3.1 PRELIMINARY: TRAINING-FREE EMBEDDING EXTRACTION

A representative attempt of MLLMs training-free embedding is E5-V (Jiang et al., 2024a). Given an input $x$ composed of arbitrary modality combinations, E5-V applies a fixed prompting template:

$$\text{"}[x] \text{ \textbackslash n Summary above content in one word:"},\tag{1}$$

and queries the MLLM to generate a single token $y$. Let $h_L(y)$ denote the hidden state of $y$ at the final transformer layer index $L$. The embedding of the input $x$ is then defined as $e(x) = h_L(y)$. This strategy preserves all parameters and thereby inherits the reasoning and generalization abilities from training. Yet its effectiveness as a generic embedder is constrained. The hidden state $h_L(y)$ is optimized for predicting the next token rather than for encoding input semantics, which biases it toward surface-level lexical statistics (Li & Zhou, 2024). Furthermore, current training-free designs often ignore the reranking stage, which is crucial for accurate retrieval.

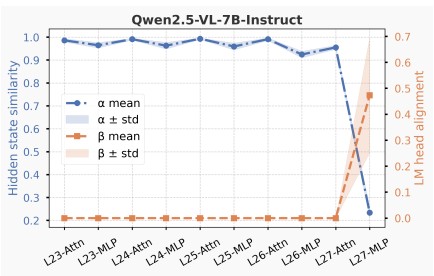 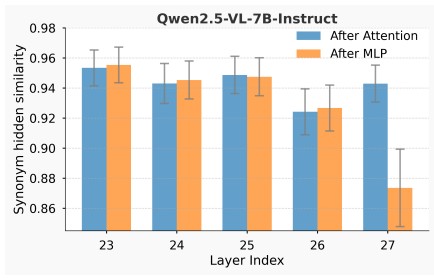

(a) Cosine similarity between adjacent layer hidden states and their alignment with LM head.

(b) Layer-wise hidden state similarity for 250 synonym pairs (mean ± standard deviation).

Figure 2: **Probing experiments on lexicalization pressure**. Results for 3B and 32B variants are provided in Fig. 7.

To overcome these issues. In § 3.2, we examine how the lexicalization pressure induced by the final MLP layer limits the semantic capacity of hidden states. In § 3.3, we move beyond one-token summarization and design a controlled generation objective. Finally, in § 3.4, we investigate the role of MLLMs as rerankers and identify a framing-effect phenomenon that undermines robustness.

### 3.2 LEXICALIZATION PRESSURE IN MLLM REPRESENTATIONS

As discussed in § 3.1, the final hidden state $h_L(y)$ is optimized for generating vocabulary logits rather than preserving semantic structure, leading to suboptimal embedding quality. Prior studies indicate that internal representations evolve across depth: early and intermediate layers capture semantic abstractions, while later layers reshape these abstractions toward task-specific objectives (Zeiler & Fergus, 2014). In MLLMs, we refer to this final transformation as *lexicalization pressure*: the process by which semantic features are projected into a space for discrete lexical prediction.

**Probing Experiment.** We begin by examining where lexicalization pressure arises within the model. Using Qwen2.5-VL at three scales (3B, 7B, and 32B), we probe the last five transformer layers. Each layer consists of an attention sub-layer and an MLP sub-layer, and we denote their outputs as $h_\ell^{\text{Attn}}$ and $h_\ell^{\text{MLP}}$ for layer index $\ell$. We assess representational shifts by the cosine similarity between consecutive sub-layer outputs: $\alpha_\ell^{\text{Attn}} = \cos\left(h_{\ell-1}^{\text{MLP}}, h_\ell^{\text{Attn}}\right)$, $\alpha_\ell^{\text{MLP}} = \cos\left(h_\ell^{\text{Attn}}, h_\ell^{\text{MLP}}\right)$. A lower $\alpha$ indicates a stronger distortion of representations. Next, we measure how strongly each hidden state is pulled into the lexical prediction space. Let $\mathbf{W} \in \mathbb{R}^{d \times |V|}$ be the LM head and let $\mathbf{w}_{y^*}$ be the column for predicted token $y^*$, we define $\beta_\ell^{\text{Attn}} = \cos\left(h_\ell^{\text{Attn}}, \mathbf{w}_{y^*}\right)$, $\beta_\ell^{\text{MLP}} = \cos\left(h_\ell^{\text{MLP}}, \mathbf{w}_{y^*}\right)$. Here, a higher $\beta$ corresponds to stronger alignment with the lexical head. The results in Fig. 2a reveal consistent trends: 1) $\alpha$ remains very high (over 0.9) across most layers but drops sharply after the final MLP (below 0.3); 2) $\beta$ stays low (around 0) across earlier layers but rises abruptly right after the last MLP (up to 0.5). These together point to *the final MLP as the focal point of lexicalization*, transforming semantically rich intermediate states into vectors aligned with token prediction.

**Effect on Semantic Retention.** As shown in Fig. 2b, cosine similarity remains around 94% across most layers, but declines to 87% after the final MLP. This suggests that lexicalization pressure compels hidden states to converge on coordinates tied to individual lexical items, erasing part of their semantic continuity. Such embeddings are therefore less suitable for retrieval tasks that require fine-grained semantic discrimination.

**Remedy.** Building on these findings, we propose a simple yet effective fix: discard the final MLP layer when producing embeddings. This choice retains the high-level abstractions encoded in deeper layers while avoiding the distortion caused by lexicalization pressure. As a result, the final representations capture semantic content more faithfully and exhibit improved robustness in retrieval tasks.

### 3.3 FROM FREE-FORM SUMMARIES TO CONTROLLED GENERATION

Early multimodal embedders (*e.g.* E5-V) often relied on simple prompting strategies such as "Summarize the input in one word". Although superficially elegant, this approach

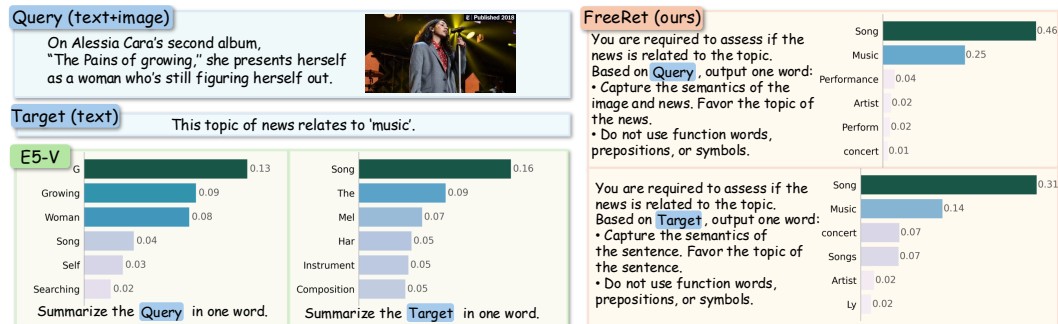

Figure 3: **Word-level probability visualization** for the output "One Word" of different methods. The top-left panel shows the input example (from N24News (Wang et al., 2021)).

leaves the generation process largely unconstrained and introduces several issues. First, *semantic loss*: compressing complex multimodal signals into a single token frequently leads to overly abstractive concepts. Second, *vocabulary noise*: high-frequency but uninformative words, including articles and prepositions, pollute the embedding space. Third, *weak task relevance*: task-agnostic prompts yield representations poorly aligned with specific retrieval needs.

Fig. 3 (left) illustrates these effects. E5-V often predicts vague words such as "Self" or "Searching", or produces spurious function words, or drifts toward semantic-related but task-irrelevant concepts like "Growing". Such outputs dilute semantic precision and lead to degraded retrieval performance.

We address these limitations by reframing free-form one-word summarization as a *controlled generation* problem. Crucially, our method requires neither architectural changes nor extra training; the improvements stem purely from prompt design. We introduce three lightweight constraints:

1) *Task alignment:* stear the generation process toward specific task priors ("You are reqired to assess if <placeholder> is related to <placeholder>.").

2) *Semantic grounding:* anchor the summary to the input concent ("Capture the semantics of <placeholder>").

3) *Noise suppression:* eliminate trivial tokens ("Do not use function words, prepositions, or symbols").

This strategy preserves the simplicity of single-word outputs yet enforces structural discipline. As shown in Fig. 3 (right), the resulting vocabularies of *Query* and *Target* are more semantically aligned with each other. Consequently, the embeddings can converge more reliably, yielding representations that remain faithful to the original content while being better tailored to the user's specific intent.

### 3.4 RERANKING WITH MLLMS: THE FRAMING EFFECT

To refine retrieval quality, we repurpose MLLMs as point-wise rerankers (Burges et al., 2005). The standard approach is straightforward: given a query-candidate pair, the model is asked to judge whether they are relevant. This forms reranking as a binary classification problem, a design choice that has become widely adopted (Zhang et al., 2025; Lin et al., 2024; Liu et al., 2025b).

Yet our study reveals that such paradigm hides surprising brittleness. The very act of framing the binary decision, even when the logical meaning remains identical, leads to strikingly different accuracies. For instance, in Fig. 4, when prompting the model with *Yes/No*, *True/False*, or *Right/Wrong*. One would expect these to be interchangeable, since each simply encodes a positive/negative decision. However, the model achieved 5.0% lower accuracy with *Right/Wrong* than with *True/False*.

What drives this sensitivity? We posit it stems from imbalances inherited from pretraining corpora. Words differ not only in logical role but also in social and pragmatic connotations: *e.g. Yes* often signals politeness, *No* conveys refusal, and *Right/Wrong* carries a moral or judgmental tone. Such contexts may bring unintended biases to their logical use. To probe this, following "context-free instruction" setup in Zhao et al. (2021), the model is prompted to choose between label pairs without any context. Ideally output logits should be uniformly distributed, but we find clear asymmetries: pairs like *Right/Wrong* or *Yes/No* show obvious skew, while *True/False* remains closer to balance. Intriguingly, greater bias correlates with lower downstream accuracy. This mirrors the classic *fram-*

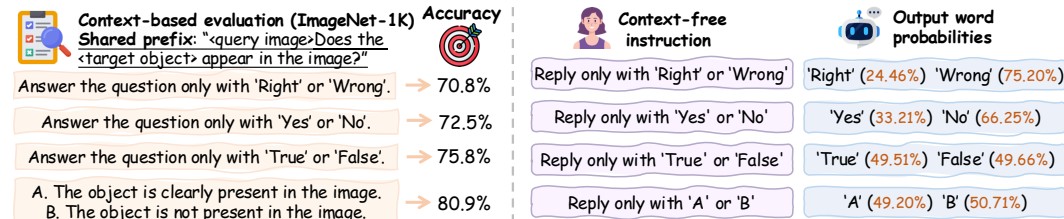

Figure 4: **LLM framing effect** on benchmark accuracy (left) and inherent lexical biases in context-free response modes (right).

*ing effect* in cognitive science, where equivalent choices elicit different judgments depending on presentation. We term the analogous phenomenon here the *LLM framing effect*.

To mitigate this effect, we frame the ranking problem as a multiple-choice question (MCQ). Given a query-candidate pair, the model is asked:

```
Task: Determine whether the candidate matches the query.
Query: {Query}
Candidate: {Candidate}
A. Yes, the candidate fully matches the query.
B. No, the candidate does not match or only partially matches.
```

The relevance score is then computed as $\texttt{SoftMax}\,(p\,(\text{`A'}))$ from the LM head. MCQ content may be adjusted to suit the needs of each task. This design offers two benefits. First, it neutralizes semantic and affective biases from lexical choices. Second, it mirrors multiple-choice formats prevalent in pretraining data, enabling more stable predictions. Despite being semantically equivalent, this simple reframing is highly effective: as shown in Fig. 4, it outperforms the commonly used *Yes/No* setup by 8.4%, underscoring the effectiveness of our design.

## 4 EXPERIMENTS

### 4.1 EVALUATION SETUP

We evaluate FreeRet on MMEB (Jiang et al., 2024b) and the video subset of MMEB-V2 (Meng et al., 2025), spanning diverse multimodal tasks: image classification (10 datasets), visual question answering (10), information retrieval (12), visual grounding (4), video classification (5), and video retrieval (5). Following prior work, we report Precision@1 for all image and video tasks.

To explore generality across model families and scales, we deploy FreeRet with the Qwen series (Qwen2-VL (Wang et al., 2024a), 2B/7B, Qwen2.5-VL (Bai et al., 2025), 3B/7B/32B, and Qwen2.5-Omni (Xu et al., 2025a)), InternVL (InternVL3 (Zhu et al., 2025), 2B/8B/14B), and LLaVA (LLaVA-OV-7B (Li et al., 2024), LLaVA-OV-1.5-8B (Contributors, 2025)).

**Baselines.** On MMEB, we compare against three groups: (1) models trained in part on the MMEB-train split (20 datasets, 662K pairs); (2) models trained on data without MMEB-train; and (3) training-free methods. For the latter, we reproduce a training-free version of E5-V with Qwen2.5-VL for fair comparison. On MMEB-V2, none of the baselines are directly supervised on the benchmark.

### 4.2 MAIN RESULTS

Table 1 and Table 2 detail FreeRet's performance on MMEB and MMEB-V2. We discuss the performance of the embedding model (FreeRet-embed) and the full retrieval pipeline (FreeRet) separately.

**Embedding Performance.** On MMEB, FreeRet-embed (w/ Qwen2.5-VL-7B) achieves an average accuracy of 53.7%, surpassing the training-free baseline E5-V by a significant margin of 13.9%. Notably, despite utilizing zero training data, FreeRet-embed proves competitive with top supervised embedders such as GME and MM-Embed. This robustness generalizes effectively to the video domain. As shown in Table 2, FreeRet-embed-2B outperforms VLM2Vec-V2-2B by 8.4% in video

Table 1: **Comparison on the MMEB benchmark** (Jiang et al., 2024b). We report Precision@1 for all models. †: we reproduce the training-free version of E5-V. Gray text denotes methods trained directly on MMEB. ‡: the amount of data used separately for the embedding and reranking models. $r_n$ denotes the reranking of the top-$n$ candidates produced by the embedding stage.

| Model | MLLM | Train Data (M) | Classification | VQA | Retrieval | Grounding | Average |
|---|---|---|---|---|---|---|---|
| *Embedding-only Methods* | | | | | | | |
| VLM2Vec (Jiang et al., 2024b) | LLaVA-1.6-7B | 0.7 | 61.2 | 49.9 | 67.4 | 86.1 | 62.9 |
| MMRet (Zhou et al., 2024) | LLaVA-1.6-7B | 26.9 | 56.0 | 57.4 | 69.9 | 83.6 | 64.1 |
| IDMR (Liu et al., 2025a) | InternVL2.5-26B | 1.2 | 66.3 | 61.9 | 71.1 | 88.6 | 69.2 |
| CAFe (Yu et al., 2025) | LLaVA-OV-7B | 0.9 | 65.2 | 65.6 | 70.0 | 91.2 | 69.8 |
| mmE5 (Chen et al., 2025) | Llama-3.2-11B-Vision | 1.2 | 67.6 | 62.6 | 71.0 | 89.6 | 69.8 |
| LLaVE (Lan et al., 2025) | LLaVA-OV-7B | 0.7 | 65.7 | 65.4 | 70.9 | 91.9 | 70.3 |
| UNITE (Kong et al., 2025) | Qwen2-VL-7B | 7.9 | 68.3 | 65.1 | 71.6 | 84.8 | 70.3 |
| UniME (Gu et al., 2025a) | LLaVA-OV-7B | 0.9 | 66.8 | 66.6 | 70.6 | 90.9 | 70.7 |
| UniME (Gu et al., 2025a) | LLaVA-1.6-7B | 0.3 | 43.0 | 17.7 | 42.5 | **63.2** | 41.6 |
| MMRet (Zhou et al., 2024) | LLaVA-1.6-7B | 26.2 | 47.2 | 18.4 | 56.5 | 62.2 | 44.0 |
| MM-Embed (Lin et al., 2024) | LLaVA-Next-7B | 1.1 | 48.1 | 32.3 | 63.8 | 57.8 | 50.0 |
| LamRA-Ret (Liu et al., 2025b) | Qwen2.5-VL-7B | 1.4 | 51.7 | 34.1 | 66.9 | 56.7 | 52.4 |
| GME (Zhang et al., 2024b) | Qwen2-VL-7B | 8.0 | 57.7 | 34.7 | 71.2 | 59.3 | 56.0 |
| E5-V† (Jiang et al., 2024a) | Qwen2.5-VL-7B | – | 41.2 | 37.2 | 37.9 | 48.4 | 39.8 |
| FreeRet-embed | LLaVA-OV-7B | – | 53.0 | 47.4 | 45.7 | 53.6 | 49.1 |
| FreeRet-embed | Qwen2-VL-7B | – | 59.0 | 50.2 | 52.3 | 60.1 | 54.5 |
| FreeRet-embed | Qwen2.5-VL-7B | – | **59.7** | **52.8** | 49.2 | 54.7 | 53.7 |
| *Embedding then Reranking* | | | | | | | |
| UniME-V2$_{r5}$ (Gu et al., 2025b) | Qwen2.5-VL-7B | 0.6 + 0.6‡ | – | – | – | – | 69.6 |
| MM-Embed$_{r10}$ (Lin et al., 2024) | LLaVA-Next-7B | 1.1 + 0‡ | 46.5 | 60.8 | 57.8 | 52.3 | 54.9 |
| LamRA$_{r10}$ (Lin et al., 2024) | Qwen2.5-VL-7B | 1.4 + 1.1‡ | 55.9 | 40.4 | 66.1 | 55.6 | 55.0 |
| FreeRet$_{r5}$ | Qwen2.5-VL-7B | – | 68.3 | 64.6 | 63.1 | 69.9 | 65.7 |
| FreeRet$_{r10}$ | Qwen2.5-VL-7B | – | 67.2 | 67.6 | 66.3 | 74.3 | 67.8 |
| FreeRet$_{r50}$ | Qwen2.5-VL-7B | – | **69.4** | **70.0** | **69.9** | **78.2** | **70.7** |

Table 2: **Comparison on two video-centric tasks** from the MMEB-V2 benchmark (Meng et al., 2025): video classification and video retrieval, each comprising five datasets. *FreeRet-embed* denotes the embedding component of FreeRet.

| Model | MLLM | Train data (M) | Video data | Video Classification | Video Retrieval |
|---|---|---|---|---|---|
| GME (Zhang et al., 2024b) | Qwen2-VL-2B | 8.0 | ✗ | 34.9 | 25.6 |
| GME (Zhang et al., 2024b) | Qwen2-VL-7B | 8.0 | ✗ | 37.4 | 28.4 |
| LamRA (Liu et al., 2025b) | Qwen2-VL-7B | 1.4 | ✗ | 39.3 | 24.3 |
| VLM2Vec (Jiang et al., 2024b) | Qwen2-VL-2B | 0.7 | ✗ | 33.4 | 20.6 |
| VLM2Vec (Jiang et al., 2024b) | Qwen2-VL-7B | 0.7 | ✗ | 39.1 | 29.0 |
| VLM2Vec-V2 (Meng et al., 2025) | Qwen2-VL-2B | 1.7 | ✓ | 39.3 | 28.8 |
| FreeRet-embed | Qwen2-VL-2B | – | – | 47.7 | 31.7 |
| FreeRet | Qwen2-VL-2B | – | – | 58.3 | 33.6 |
| FreeRet-embed | Qwen2-VL-7B | – | – | 54.3 | 36.5 |
| FreeRet | Qwen2-VL-7B | – | – | **63.2** | **39.3** |

classification and 2.9% in video retrieval. These results demonstrate that our controlled-generation strategy successfully mitigates lexicalization pressure and generalizes across modalities without the need for the extensive in-domain fine-tuning required by prior approaches.

**Reranking Performance.** Incorporating the reranking stage yields substantial additional gains. On MMEB, the full FreeRet pipeline attains an average accuracy of 70.7% ($r50$), outperforming strong two-stage baselines like MM-Embed and LamRA by approximately 13%. This margin is particularly significant given that these baselines are fine-tuned on millions of multimodal pairs. Furthermore, FreeRet matches the performance of UniME-V2 (69.6%), a model trained directly on MMEB with advanced optimization techniques. Similar trends are observed in the video tasks, where the addition of reranking further solidifies the lead of FreeRet over supervised counterparts.

Overall, these results highlight two key insights: (i) post-training approaches rely heavily on supervised data and often exhibit limited generalization under domain or modality shifts; (ii) in contrast, FreeRet exhibits robust zero-training performance across datasets, tasks, and modalities, establishing a new standard for training-free multimodal retrieval.

### 4.3 ABLATION STUDY

**Effect of the Final MLP Layer.** We first probe the role of the final MLP layer, which we posit introduces lexicalization pressure that degrades embedding quality. As a baseline, we adopt the hidden state from the last transformer layer ($h_L^{\mathrm{MLP}}$), a standard choice in prior work. We then ablate at three depths: (i) bypassing the final MLP by using $h_L^{\mathrm{Attn}}$, (ii) removing the full last transformer layer by using $h_{L-1}^{\mathrm{MLP}}$, and (iii) removing the last two transformer layers by using $h_{L-2}^{\mathrm{MLP}}$. Results

Table 3: **Ablation study of FreeRet** with Qwen2.5-VL (3B and 7B). Results are reported as Precision@1, averaged over 36 MMEB datasets. The baseline configuration is highlighted in gray, while the FreeRet configuration is shown in green.

(a) Alleviate lexicalization pressure.

| Embedding | 3B | 7B |
|---|---|---|
| $h_L^{\mathrm{MLP}}$ | 45.34 | 47.97 |
| $h_L^{\mathrm{Attn}}$ | 50.67 | **53.68** |
| $h_{L-1}^{\mathrm{MLP}}$ | **51.04** | 51.03 |
| $h_{L-2}^{\mathrm{MLP}}$ | 50.64 | 48.78 |

(b) Control embedding generation.

| Configuration | 3B | 7B |
|---|---|---|
| (a): base instruct Eq. (1) | 42.42 | 45.53 |
| (b): (a) + semantic ground | 46.71 | 50.60 |
| (c): (b) + noise suppress | 48.20 | 51.50 |
| (d): (c) + task align | **50.67** | **53.68** |

(c) Neutralize LLM framing effect.

| Label framing | 3B | 7B |
|---|---|---|
| Right-Wrong | 59.46 | 64.71 |
| Yes-No | 58.39 | 65.28 |
| True-False | 60.06 | 66.71 |
| Multiple-choice | **60.31** | **70.72** |

in Tab. 3a show that eliminating only the final MLP yields consistent gains: improving the 3B and 7B models by 5.33% and 5.71%, confirming that lexicalization pressure indeed harms representation quality. However, discarding additional layers fails to bring further benefits; the 7B variant even degrades. We hypothesize that the deeper layers capture essential semantic abstractions. The effect is magnified in shallower architectures such as Qwen2.5-VL 7B (28 layers) compared to the 3B variant (36 layers).

**Controlling Embeddings via Prompts.** We then investigate whether embeddings can be guided at inference without altering parameters. Starting from a simple instruction (Eq. (1)), we progressively add lightweight constraints (see Tab. 3b). Explicit semantic grounding substantially improves alignment, yielding 4.29% and 5.07% gains for 3B and 7B models. Adding noise-suppression instructions further attenuates spurious function words, giving another increase of 1.49% and 0.9%. Finally, encoding task-specific priors yields an additional boost of 2.47% and 2.17%. Together, these results demonstrate that the embedding space of large models can be effectively reshaped through prompting control, offering a parameter-free avenue for fine-grained representation steering.

**Study on the LLM Framing Effect.** Next, we test whether the surface framing of reranking questions impacts performance. As shown in Tab. 3c, semantically neutral labels such as *True/False* consistently outperform alternatives with that carry social or pragmatic connotations, such as *Yes/No* or *Right/Wrong*. Our multiple-choice framing achieves the highest accuracy, due to its neutrality as well as consistency with pretraining distributions, where such formats are frequent. Interestingly, sensitivity increases with model scale. The 7B variant shows large variance across framings, while the 3B model remains relatively stable. We conjecture that larger models capture finer semantic distinctions and are thus more vulnerable to subtle framing shifts, whereas smaller models operate on coarser abstractions less affected by such perturbations.

**Impact of the Reranking Stage.** Finally, we examine the contribution of our reranking stage. Prior multimodal retrievers typically optimize the embedding model while underexploring reranking, yet accurate candidate selection heavily depends on this step. In Fig. 5, we vary the number of candidates passed to the reranking stage.

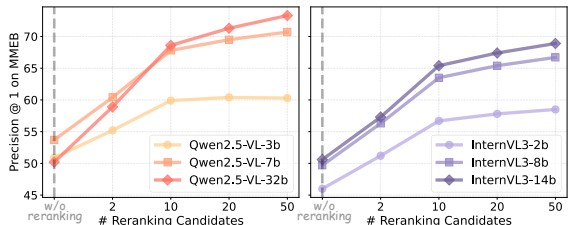

Figure 5: Varying the number of reranking candidates.

Benchmark performance consistently improves as the reranking pool enlarges, underlining its necessity. Notably, our FreeRet reuses the same model across both embedding and reranking, avoiding additional parameterization or deployment overhead. This yields a favorable balance between precision and efficiency, positioning reranking as an indispensable yet economical design choice.

## 4.4 DISCUSSIONS ON TRAINING-FREE ADVANTAGES

**Instant Deployment.** A key strength of the training-free paradigm is its ability to turn any MLLM into a retriever immediately, with no additional fine-tuning. This property allows practitioners to flexibly leverage diverse model families and scales depending on task requirements. Given the rapid pace of new MLLM releases, the cost of repeated fine-tuning quickly becomes prohibitive. By eliminating this overhead, our approach enables faster adoption of new advances. Moreover, the training-free nature broadens MLLM evaluation beyond conventional QA objectives to include retrieval-oriented benchmarks, offering a richer assessment of their multimodal reasoning capacity.

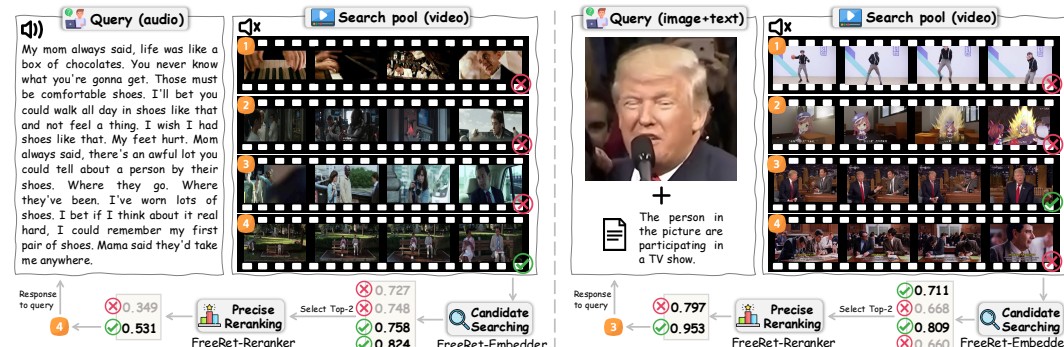

Figure 6: **FreeRet enables instant omni-modal retrieval** with omni-modality models. Illustrated with Qwen2.5-Omni: audio-to-video retrieval (left); image+text to video retrieval (right).

**Scalable Omni-Modality Retrieval.** Training-based multimodal retrieval encounters a fundamental scalability bottleneck: each new modality demands extensive paired data covering all query–target configurations. The data requirement grows combinatorially. For example, with four modalities (text, image, video, audio), there exist $\sum_{i=1}^{4} \binom{4}{i} = 15$ possible modality combinations, leading to $15 \times 15 = 225$ query–target pair types. This cost escalates further as modalities increase, rendering full coverage infeasible in practice. Our method sidesteps this limitation by directly exploiting the intrinsic omni-modal understanding already embedded in MLLMs. As shown in Fig. 6, FreeRet supports retrieval across arbitrary configurations, *e.g.*, retrieving a video using an audio query or a joint image–text query, even when the model has never been trained on such pairs. In this way, omni-modality retrieval shifts from a data-intensive challenge to a natural emergent capability.

**Preserving Multimodal Intelligence.** Unlike fine-tuning pipelines that risk eroding a model's pretrained strengths, FreeRet preserves the multimodal reasoning and conversational abilities of the underlying MLLM. Retrieval is introduced as an additional functionality, not as a replacement. Consequently, the pretrained capabilities remain intact while retrieval emerges as a first-class operation. This design enables a single model to natively support retrieval, reranking, and generation within a unified RAG framework, avoiding the fragmentation introduced by specialized expert modules. The result is both a reduction in engineering complexity and an increase in deployment efficiency.

**Towards Long-Video Understanding.** Reasoning over long videos is particularly challenging due to extended temporal contexts, where uniform frame sampling often dilutes attention with redundant content. FreeRet offers a pragmatic solution: it first retrieves the most relevant frames, thereby grounding subsequent reasoning on evidence-rich content. This retrieval-driven focus effectively reduces temporal redundancy. On the hour-level

Table 4: **Performance gains on LVBench**: Qwen2.5-VL with FreeRet.

| Sample Method | # Frames | LVBench |
|---|---|---|
| Uniform | 64 | 39.0 |
| FreeRet | 64 | 44.8 (**+5.8**) |
| Uniform | 32 | 39.0 |
| FreeRet | 32 | 44.2 (**+5.2**) |
| Uniform | 16 | 36.7 |
| FreeRet | 16 | 42.7 (**+6.0**) |

benchmark LVBench (Wang et al., 2024b), experiments with Qwen2.5-VL 7B (see Tab. 4) demonstrate consistent improvements. These results highlight the promise of FreeRet as a foundation for scaling MLLMs toward long-horizon multimodal reasoning.

## 5 CONCLUSION

This work demonstrates that pretrained MLLMs can act as effective retrieval engines without any additional training. By decoupling embedding extraction from lexical alignment, conditioning representation with explicit priors, and neutralizing framing in reranking, FreeRet turns off-the-shelf MLLMs into strong two-stage retrievers. Experiments on MMEB show that FreeRet not only surpasses heavily trained baselines but also remains competitive with MMEB-supervised methods. Beyond performance, its plug-and-play nature preserves reasoning ability, supports arbitrary modality combinations, and integrates retrieval with generation in a single model. These results challenge the prevailing reliance on costly contrastive training and point toward a retrieval paradigm where generalist MLLMs serve as unified, training-free backbones for multimodal reasoning and generation.

## 6 REPRODUCIBILITY STATEMENT

FreeRet is implemented within a widely adopted search-then-reranking retrieval framework. In the search stage, we show our prompting strategies in § 3.3 and Fig. 3, and extract features following the **Remedy** procedure described in § 3.2. For the reranking stage, our process is detailed in § 3.4, covering both multiple-choice question framing and score computation. To further support reproducibility, we provide runnable code in the supplemental material.

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

# A APPENDIX

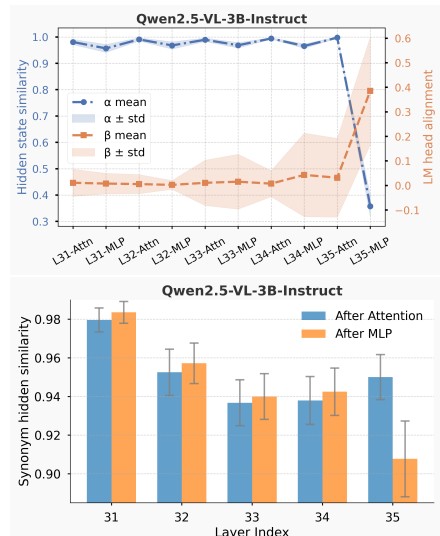 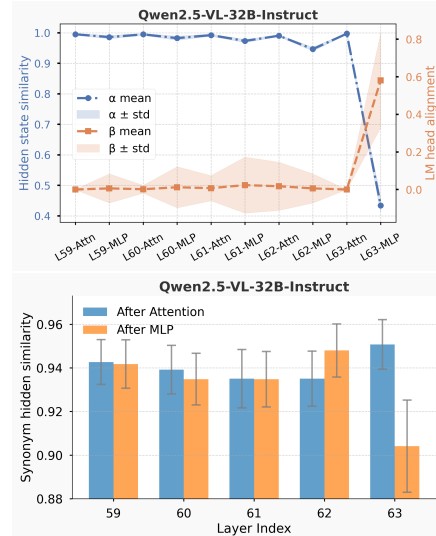

Figure 7: Qwen2.5-VL 3B and 32B results in probing experiments on lexicalization pressure.

Table 5: Ablate the choice of removing the final MLP layer on models beyond Qwen. We report average Precision@1 scores on the MMEB benchmark.

| Embedding | LLaVA-OV-1.5-8B | InternVL3-2B | InternVL3-8B |
|---|---|---|---|
| $h_L^{\text{MLP}}$ | 47.8 | 40.2 | 44.2 |
| $h_L^{\text{Attn}}$ | 54.2 (+6.4) | 46.0 (+5.8) | 49.7 (+5.5) |

## A.1 ADDITIONAL DETAILS ON LEXICALIZATION PRESSURE EXPERIMENTS

We investigate the locus and consequences of lexicalization in multi-modal large language models through probing experiments across different model scales. In the main text, we report results on Qwen2.5-VL 7B, showing that lexicalization is strongly concentrated in the final MLP layers. This concentration appears to sharpen token-level alignment, but it comes at the cost of semantic coherence, suggesting that lexicalization exerts a structural pressure on representation quality.

To validate the robustness of this finding, we further evaluate Qwen2.5-VL 3B and 32B. As shown in Fig. 7, all models exhibit consistent patterns: the final MLP absorbs the majority of lexicalization load and induces similar trade-offs between lexical grounding and semantic fidelity. This convergence across scales indicates that lexicalization pressure is not an artifact of model size, but instead emerges as a general structural property of the architecture.

Our evaluation involves 720 data points, sampled systematically from MMEB. Specifically, we select 20 examples from each of the 36 datasets. For all reported scores, we compute both the mean and the standard deviation, ensuring the conclusions are driven by stable trends rather than dataset-specific artifacts.

Moreover, we extend our analysis beyond the Qwen series by evaluating additional architectures, including InternVL and LLaVA. The results, summarized in Tab. 5, demonstrate that lexicalization pressure is a pervasive phenomenon. Importantly, our simple modification (removing the final MLP layer) consistently improves performance across diverse MLLM architectures.

To further substantiate the claim that semantic resolution degrades after the final MLP layer, we analyze hidden-state similarities for *synonym*, *antonym*, and *random word pairs* across every attention and MLP block within the last five transformer layers. To ensure that the evaluation genuinely reflects semantic content rather than superficial lexical overlap, we employ cross-lingual English–Chinese word pairs with matched meaning, opposite meaning, or no semantic relation. As shown in Fig. 8, representations before last MLP preserve clear semantic structure: synonym pairs exhibit the highest similarity; antonym pairs are only slightly lower, likely due to antonyms typically

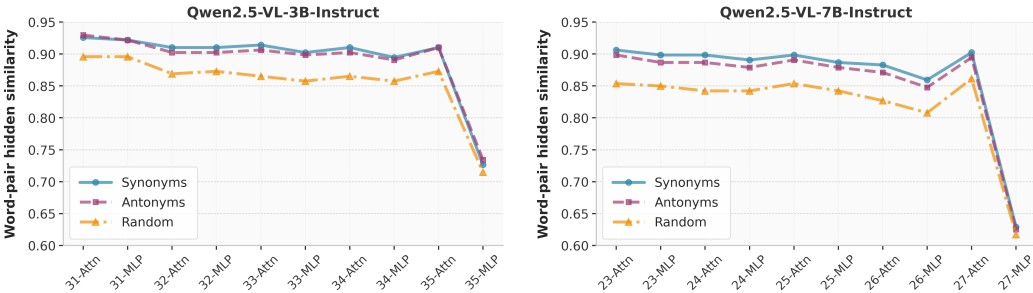

Figure 8: Hidden-state similarity of synonym, antonym, and random word pairs across attention and MLP blocks in the final five transformer layers.

share same domain, and LLMs trained without contrastive objectives tend to cluster domain-level meanings even when the words express opposite concepts; random word pairs show the lowest similarity, as expected. However, immediately after the final MLP layer, the gap among these three categories collapses, with all three categories converging to nearly identical similarity levels. This sharp loss of semantic separation indicates that the final MLP layer overrides upstream semantic organization, providing direct evidence that lexicalization pressure deteriorates semantic representation quality. The phenomenon aligns with the downstream performance degradation reported in Tab. 3a.

## A.2 ADDITIONAL THEORETICAL ANALYSIS

We provide a simplified argument illustrating why discarding the final MLP layer can preserve semantic fidelity of hidden states.

**Lemma 1** (Lexicalization Alignment). *Let $h \in \mathbb{R}^d$ be the hidden state before the last MLP, and define*

$$h' = Ah + b, \quad A \in \mathbb{R}^{d \times d}, \ b \in \mathbb{R}^d.$$

*Let $\mathbf{W} = [\mathbf{w}_1, \ldots, \mathbf{w}_{|V|}] \in \mathbb{R}^{d \times |V|}$ be the LM head and*

$$\mathcal{L}(h') = -\log \frac{\exp(\mathbf{w}_{y^*}^\top h')}{\sum_{v \in V} \exp(\mathbf{w}_v^\top h')}$$

*the cross-entropy loss. Then the gradient is*

$$\nabla_{h'}\mathcal{L} = \sum_{v \in V} p(v|h')\mathbf{w}_v - \mathbf{w}_{y^*},$$

*which points toward aligning $h'$ with $\mathbf{w}_{y^*}$ while suppressing components orthogonal to* $\mathrm{span}(\mathbf{W})$.

*Proof.* Since $\nabla_{h'}\mathcal{L}$ is always a linear combination of vocabulary vectors $\{\mathbf{w}_v\}$, optimization over $\mathcal{L}$ depends only on the projection of $h'$ onto $\mathcal{W} = \mathrm{span}\{\mathbf{w}_v\}$. Any orthogonal component $\epsilon = h' - P_{\mathcal{W}}(h')$ satisfies $\nabla_{h'}\mathcal{L}^\top \epsilon = 0$ and is thus uncontrolled by the objective. Training therefore drives $h'$ to maximize $\langle h', \mathbf{w}_{y^*} \rangle$ while diminishing the effective role of $\epsilon$. $\square$

**Corollary A.0.1** (Lexicalization Pressure). *The final MLP layer learns $h' \approx P_{\mathcal{W}}(h)$, which increases alignment with the lexical head but reduces retention of semantic features outside $\mathcal{W}$.*

*Remark* (Practical Implication). By discarding the final MLP when producing embeddings, we bypass the forced projection into $\mathcal{W}$, thereby retaining semantic components of $h$ that would otherwise be suppressed by lexicalization pressure. This aligns with our empirical findings in Fig. 2b.

## A.3 ADDITIONAL DETAILS ON WORD-LEVEL PROBABILITY VISUALIZATION

To better understand the representational behavior of our model, we visualize word-level probabilities as shown in Fig. 3. Although our main method removes the final MLP layer to ensure more faithful embeddings, we reintroduce this layer solely for visualization purposes. Specifically, we

pass the hidden states through the original language modeling head followed by a softmax, which produces interpretable token-wise probabilities.

This procedure allows us to probe the model's internal distribution over the vocabulary without altering the underlying training or inference pipeline. Compared with E5V, the key difference lies in our prompt design, which directly shapes the representation generation process. This design choice provides a clearer window into how our approach influences semantic alignment, highlighting the improvements our method achieves over prior baselines.

### A.4 ADDITIONAL DETAILS ON LLM FRAMING EFFECT EXPERIMENTS

For benchmark evaluation, we employ the ImageNet-1K (Deng et al., 2009) subset from the MMEB benchmark. To ensure robustness, we rephrase the shared prefix (e.g., the prompt question) three times and report the average accuracy across these variants.

For the context-free instruction setting, we further mitigate position-related biases by swapping the order of the labels. For instance, we alternate between instructions such as "`Reply only with 'Yes' or 'No'`" and "`Reply only with 'No' or 'Yes'`". We then average the corresponding logits to minimize the influence of positional effects in the instruction text.

### A.5 THE POTENTIAL OF FREERET IN MASSIVE-CANDIDATE SCENARIOS

In large-scale retrieval, inference efficiency is often as critical as accuracy. While FreeRet, built upon modern MLLMs, achieves strong performance, it incurs higher latency compared to CLIP-based methods, a limitation also noted in prior MLLM-based retrievers (Lin et al., 2024; Liu et al., 2025b; Cui et al., 2025).

To ensure scalability in scenarios with over 100M candidates (a setting relevant to many real-world applications), we propose three orthogonal strategies, readily applicable to FreeRet and other MLLM-based retrievers:

1. **Coarse pre-filtering.** Employing an extremely lightweight model for initial filtering reduces the effective candidate pool before invoking MLLM-based retrieval and reranking.

2. **Controlled reranking.** Since reranking dominates inference cost, limiting the number of reranked candidates yields significant efficiency gains with minimal performance loss (see Fig. 5).

3. **Lightweight MLLMs.** Substituting the backbone with more efficient MLLMs provides further savings, and ongoing progress in model compression suggests increasingly favorable trade-offs in the near future.

### A.6 ERROR ANALYSIS

As illustrated in Fig. 9, our model exhibits two primary failure modes.

First, the model tends to over-emphasize explicit object semantics while under-utilizing implicit high-order relational cues. In the top example of Fig. 9(a), although the model correctly identifies both the cat and the food bowl, it incorrectly predicts the action "eat," failing to interpret the cat's upward-facing, crouched posture that signals an imminent jump. A similar issue appears in the bottom example: despite detecting both the cake and the candle, the model overlooks the child's focused gaze on the candle flame, causing it to choose "cake" instead of the ground-truth answer "candle."

Second, we observe a bias in the image–text fusion process, where the model overly relies on visual similarity at the expense of faithfully following the textual instruction. In the bottom example of Fig. 9(b), the query specifies "three bottles of soft drink," yet the model selects an image containing only one bottle because it is visually closer to the query. In the top example, the presence of the original query image within the retrieval pool amplifies this bias: the visually identical but semantically incorrect candidate ends up dominating the model's decision.

Together, these findings may offer insights for the future development of training-free retrievers.

(a) Implicit relational cues overlooked       (b) Multimodal fusion bias toward visual similarity

Figure 9: Representative failure cases of our model. (a) Errors arising from missed implicit relational cues: the model focuses on explicit object semantics while overlooking posture, gaze, and other high-order relations. (b) Errors caused by multimodal fusion bias: the model over-prioritizes visual similarity and fails to strictly follow textual instructions, leading to incorrect retrieval despite semantically precise prompts.

Table 6: **FreeRet generalizes well across model families and scales.**

| Model | MLLM | Classification | VQA | Retrieval | Grounding | Average |
|-------|------|----------------|-----|-----------|-----------|---------|
| FreeRet | LLaVA-OV-7B | 59.6 | 58.7 | 62.4 | 65.9 | 61.0 |
| FreeRet | LLaVA-OV-1.5-8B | 62.8 | 69.2 | 65.9 | 65.3 | 65.9 |
| FreeRet | InternVL3-2B | 59.1 | 58.2 | 56.2 | 65.1 | 58.5 |
| FreeRet | InternVL3-8B | 62.3 | 68.9 | 64.1 | 79.9 | 66.7 |
| FreeRet | InternVL3-14B | 61.1 | 71.2 | 67.9 | 85.1 | 68.9 |
| FreeRet | Qwen2-VL-7B | 65.6 | 64.1 | 68.7 | 75.2 | 67.2 |
| FreeRet | Qwen2.5-VL-3B | 57.5 | 61.2 | 62.8 | 57.8 | 60.3 |
| FreeRet | Qwen2.5-VL-7B | 69.4 | 70.0 | 69.9 | 78.2 | 70.7 |
| FreeRet | Qwen2.5-VL-32B | 70.8 | 75.4 | 72.4 | 77.0 | 73.3 |

## A.7 GENERALIZATION ACROSS MODEL FAMILIES AND SCALES

FreeRet consistently delivers superior multimodal retrieval performance across diverse model families and parameter scales, as shown in Tab. 6. Applied to LLaVA, InternVL, and Qwen-VL variants, FreeRet yields robust gains in classification, VQA, retrieval, and grounding. Larger backbone models tend to benefit more, with Qwen2.5-VL-32B achieving the highest average score (73.3). The strong grounding performance of InternVL3-14B (85.1) further highlights FreeRet's ability to leverage InternVL3's spatially grounded reasoning. Overall, these results indicate that FreeRet generalizes well across architectures and scales, providing consistent improvements without any fine-tuning.

## A.8 COMPUTATIONAL EFFICIENCY ANALYSIS

Table 7: **Comparison of computational efficiency with existing two-stage retrievers.**

| Method | Foundation (embed+rerank) | Unify | Train Data | GPU Memory (embed+rerank) | Latency (embed) | Precision@1 (embed) | Latency (embed+rerank) | Precision@1 (embed+rerank) |
|--------|---------------------------|-------|------------|---------------------------|-----------------|---------------------|------------------------|----------------------------|
| MM-Embed | NV-Embed-7B + LLaVA-Next-7B | ✗ | 1.1M | 33.28 GB | 0.11s | 52.8 | 1.51s | 54.9 |
| LamRA | Qwen2.5VL-7B + Qwen2.5VL-7B | ✗ | 1.4M | 30.91 GB | 0.08s | 51.3 | 0.78s | 55.0 |
| FreeRet | Qwen2.5VL-3B | ✓ | - | 9.01 GB | 0.08s | 50.9 | 0.35s | 59.9 |
| FreeRet | Qwen2.5VL-7B | ✓ | - | 17.75 GB | 0.08s | 53.7 | 0.47s | 67.8 |

Compared with the embedding-only pipeline, adding a reranking stage naturally increases inference-time computational cost. To characterize this overhead, we analyze the efficiency of FreeRet by (1) comparing it with previous two-stage retrievers and (2) examining how performance and resource usage vary with the number of reranking candidates. In both evaluations, we run each method across the entire MMEB benchmark, measuring only the model's forward pass and reporting average per-sample latency and peak GPU memory usage.

Tab. 7 summarizes the comparison with prior two-stage retrievers. FreeRet achieves substantially better end-to-end retrieval performance while being far more efficient. Because existing methods rely on separate embedding and reranking models, their memory footprint is nearly doubled. In

Table 8: **Computational efficiency analysis with different numbers of reranking candidates.**

| Backbone | # Candidate | Latency (s) | GPU Memory (GB) | MMEB Precision@1 |
|---|---|---|---|---|
| | 0 | 0.08 | 7.09 | 50.9 |
| | 2 | 0.16 | 8.36 | 55.2 |
| Qwen2.5VL 3B | 10 | 0.35 | 9.01 | 59.9 |
| | 20 | 0.81 | 9.94 | 60.4 |
| | 50 | 2.12 | 12.97 | 60.3 |
| | 0 | 0.08 | 14.58 | 53.7 |
| | 2 | 0.21 | 16.76 | 60.4 |
| Qwen2.5VL 7B | 10 | 0.47 | 17.75 | 67.8 |
| | 20 | 1.06 | 19.13 | 69.5 |
| | 50 | 2.97 | 23.59 | 70.7 |

contrast, FreeRet unifies embedding and reranking within a single MLLM. This architecture enables the reranking stage to reuse the vision-encoder outputs and the shared-prefix KV cache, greatly reducing computational overhead.

Tab. 8 reports efficiency under different reranking candidate sizes. Without reranking, FreeRet maintains efficiency comparable to standard embedding-only systems. As the number of candidates increases, both latency and memory usage grow accordingly. For practical deployment, the number of candidates offers a flexible knob to balance efficiency and effectiveness, enabling applications to choose the optimal trade-off for their resource budgets.

### A.9    LIMITATIONS

As discussed in § A.5, FreeRet inherits the computational overhead of MLLM-based retrievers, which may limit its practicality in resource-constrained environments. Furthermore, unlike data-driven methods that can adapt through task-specific training, FreeRet is entirely training-free and relies solely on the underlying multimodal understanding and instruction-following capability of the foundation model. Consequently, its performance may degrade when the base model itself provides weak representations or multimodal reasoning ability.

### A.10    CODE DEMO

To facilitate understanding of both the implementation and the results of our proposed FreeRet, we provide a demo code package. The current demo supports Qwen2.5-VL, with a reference implementation for the image+text to image retrieval setting. More generally, it can handle arbitrary combinations of image, text, and video modalities for both queries and targets. Furthermore, the framework can readily extend to additional modalities when paired with omni-modal models such as Qwen2.5-Omni. We encourage the reviewers to experiment with the demo to better appreciate the flexibility and practicality of our approach.

### A.11    THE USE OF LARGE LANGUAGE MODELS

In this work, Large Language Models were used exclusively to refine and improve the clarity of our writing.

