# OpenReview forum: "FreeRet: MLLMs as Training-Free Retrievers"
_ICLR.cc/2026/Conference — Submitted to ICLR 2026_

### Official Review · Reviewer_e7fh · 2025-10-29

**Soundness:** 4
**Presentation:** 3
**Contribution:** 3
**Rating:** 8
**Confidence:** 5

**Summary:**

This paper presents FreeRet, a two-stage framework that directly uses off-the-shelf MLLMs as "training-free" retrievers: first, use the internal representations of the model for candidate recall, and then use the same model for fine-grained re-ranking. The core idea is to no longer perform contrastive learning or additional supervision, but rather to transform the model's semantic understanding and reasoning abilities into usable retrieval signals through careful design of architecture and prompts. The three key technical points of the authors are as follows:
- Semantic Representation Extraction: Bypass the last layer MLP before the language head to alleviate "lexicalization pressure", and use more semantically faithful hidden states as embeddings;
- Controlled Representation Generation: Using prompts that incorporate task priors, semantic anchoring, and noise suppression, the generation process of "summarize in one word" is constrained to be a more stable and topic-relevant representation;
- Neutralization rearrangement hint: Rewrite the binary classification relevance judgment into a multiple-select question to alleviate the bias of labels such as Yes/No in pre-trained corpora (which the author refers to as the "LLM framing effect").

On 46 datasets such as MMEB and MMEB-V2, FreeRet significantly outperformed multiple comparison methods trained with millions of samples without any additional training; under larger model scales, it even approached or exceeded the SOTA directly supervised on the benchmark.

**Strengths:**

The innovation of the paper does not lie in inventing a brand-new architecture, but in proposing three key and mutually coordinated approaches to "how to transform the potential capabilities of existing MLLMs into high-quality retrieval signals": First,  analyze and avoid the lexicalization pressure of the last layer of MLP , supported by probe experiments on intra-layer similarity and alignment with the LM head; Second,  transform the freely generated one-word summary into controlled generation , constrained by concise prompts to align the vocabulary space with the task; Third,  reveal and mitigate the pragmatic bias of re-ranking labels , replacing binary labels with MCQ to obtain more stable judgments. These all belong to the originality of "de-engineering": significantly improving the usability of representation and discrimination without modifying the model or conducting training.

**Weaknesses:**

1. The measurement of the framing effect is mainly conducted in classification scenarios, and it is recommended  to extend it to VQA, retrieval question answering, and cross-modal attribute matching scenarios  to examine the sensitivity of MCQ copywriting design to different tasks.

2. The paper discusses engineering strategies under ultra-large candidate scales in the appendix, but  measured overhead figures  (throughput, latency, and video memory usage) are still lacking. Suggestions:
- Under unified hardware and batch settings, report the end-to-end latency and GPU utilization of recall and re-ranking;
- Presentthe curve of candidate pool size - performance - latencyto quantify the cost - effectiveness inflection point of "controlled reordering candidate count";

3. The author emphasizes the cross-family and cross-task stability and suggests further:
- Supplementary Error Analysis : Statistically analyze the most easily confused negative example types and error causes (such as lexical bias, modal alignment failure, long-distance dependency loss, etc.) to guide subsequent prompt or structural improvements;
- More experiments on RAG task such as WebQA and MultimodalQA.
- More experiments on retrieval task such as Flickr30K and MS-COCO.

**Questions:**

1. Although controlled prompts suppress function words, the differences in common word frequencies and morphological structures among different languages are relatively large. Have the authors considered generating within a restricted task vocabulary/category set (e.g., news topic sets, object category sets) to avoid cross-lingual word distribution drift? If run on Chinese or mixed corpora, what impact does lemmatization have on performance?

2. In VQA, cross-modal attribute matching, or retrieval question answering, the  option text  and  explanatory prompt  of MCQ may affect the coupling of the model's "reasoning - judgment". Have the authors systematically compared the effects of different MCQ copywriting (more neutral vs. more fine grain)? Is there an inflated result caused by "overly strong prompts"?

3. Though the paper mentions that framing the ranking problem as a MCQ is to alleviate the LLM framing effect, it is mentioned in [1] that the responses of LLMs to MCQ are actually unstable. How do the authors view this issue?

ref:

[1] Zheng C, Zhou H, Meng F, et al. Large language models are not robust multiple choice selectors[J]. arXiv preprint arXiv:2309.03882, 2023.

---

> ### Author Response · Authors · 2025-11-23
>
> > **Q1: The measurement of the framing effect is mainly conducted in classification scenarios, and it is recommended to extend it to VQA, retrieval question answering, and cross-modal attribute matching scenarios to examine the sensitivity of MCQ copywriting design to different tasks.**
>
> Thank you for your constructive suggestions.
>
> Our measurement of the framing effect is conducted from two complementary perspectives: (1) **context‑free lexical bias**, which is task‑agnostic, and (2) **benchmark accuracy**, which reflects task‑specific sensitivity. While Fig. 4 focuses on classification for ease of presentation, Tab. 3(c) reports averaged performance across all 36 MMEB datasets to provide a task‑comprehensive view.
>
> Following your suggestion, we further break down the analysis across the four task types you mentioned. As shown below, classification and visual grounding exhibit noticeably larger variance across framing styles, whereas retrieval remains relatively stable.
>
> |             | Classification | VQA  | Retrieval | Visual Grounding | Average |
> | ----------- | -------------- | ---- | --------- | ---------------- | ------- |
> | Right/Wrong | 60.4           | 65.2 | 66.6      | 68.6             | 64.7    |
> | Yes/No      | 61.3           | 66.4 | 67.1      | 67.3             | 65.3    |
> | True/False  | 63.6           | 67.4 | 67.2      | 71.5             | 66.7    |
> | MCQ         | 69.4           | 70.0 | 69.9      | 78.2             | 70.7    |
>
> > **Q2: Measured overhead figures (throughput, latency, and memory usage) are still lacking. Suggestions: 1) Under unified hardware and batch settings, report the end-to-end latency and GPU utilization of recall and re-ranking; 2) Presentthe curve of candidate pool size - performance - latency to quantify the cost-effectiveness inflection point of "controlled reordering candidate count".**
>
> Thank you for the constructive suggestions.
>
> For the first point, please refer to our response to **Q3 of Reviewer 5R9b**. There, under unified hardware (one A800‑80GB) and batch size 1, we report per‑sample latency and peak GPU memory for (i) recall‑only and (ii) the full recall‑then‑reranking pipeline, comparing our FreeRet with existing retriever baselines.
>
> For the second point, Fig. 5 already visualizes the relationship between candidate pool size and downstream performance. To more fully characterize the cost–effectiveness trade‑off, following your suggestions, we additionally provide per-sample latency, GPU memory, and MMEB Precision@1 across candidate sizes:
>
> | Backbone     | # Candidate | Latency (s) | GPU Memory (GB) | Precision@1 |
> | ------------ | ----------- | ----------- | --------------- | ----------- |
> | Qwen2.5VL 3B | 0           | 0.08        | 7.09            | 50.9        |
> | Qwen2.5VL 3B | 2           | 0.16        | 8.36            | 55.2        |
> | Qwen2.5VL 3B | 10          | 0.35        | 9.01            | 59.9        |
> | Qwen2.5VL 3B | 20          | 0.81        | 9.94            | 60.4        |
> | Qwen2.5VL 3B | 50          | 2.12        | 12.97           | 60.3        |
> | Qwen2.5VL 7B | 0           | 0.08        | 14.58           | 53.7        |
> | Qwen2.5VL 7B | 2           | 0.21        | 16.76           | 60.4        |
> | Qwen2.5VL 7B | 10          | 0.47        | 17.75           | 67.8        |
> | Qwen2.5VL 7B | 20          | 1.06        | 19.13           | 69.5        |
> | Qwen2.5VL 7B | 50          | 2.97        | 23.59           | 70.7        |
>
> We will plot these numbers as a curve and include it in the revised paper.
>
> > **Q3: Analyze the most easily confused negative example types and error causes.**
>
> Thank you for your suggestions.
>
> After manually reviewing several hundred error cases across multiple datasets, we observed two recurring error patterns in FreeRet. First, the model tends to over‑emphasize explicit object semantics while under‑utilizing higher‑order relational and contextual cues. Second, the model overly relies on surface‑level visual similarity, at times failing to fully adhere to the textual instruction. **Representative examples and analyses have been added to Sec. A.6 of the revised paper.**
>
> In summary, these error types suggest that the model depends more on surface‑level perceptual matching than on deeper reasoning. Incorporating stronger reasoning mechanisms (particularly for complex user intents and heterogeneous modality combinations) presents a valuable direction for future improvement.

---

> ### Author Response · Authors · 2025-11-23
>
> > **Q4: More experiments on RAG task (WebQA, MultimodalQA), and retrieval task (Flickr30K, COCO).**
>
> Thank you for the valuable suggestions.
>
> For RAG tasks, WebQA is already included in MMEB, which we test in our paper. In addition, we provided results on the extreme long‑video benchmark LVBench (Tab. 4) using **video RAG** technique, where our retrieved frames substantially outperform uniform frame sampling. We agree that broader RAG evaluations are beneficial and will include them in future work.
>
> For Flickr30K and MS‑COCO, we evaluate FreeRet, FreeRet‑embed (FreeRet w/o reranking), and two representative baselines: E5‑V (training‑free varient) and VLM2Vec (training‑based; optimized on MMEB-662K which includes COCO).
>
> The results are shown below. Even with a 3B MLLM, FreeRet‑embed significantly surpasses E5‑V across both datasets. Adding the reranking stage leads to large further gains, achieving performance comparable to VLM2Vec despite requiring no training.
>
> We will incorporate these experiments in the revised version.
>
> | Image Retrieval         | Train data | MLLM          | Flickr30K R@1 | Flickr30K R@5 | Flickr30K R@10 | COCO R@1 | COCO R@5 | COCO R@10 |
> | ----------------------- | ---------- | ------------- | ------------- | ------------- | -------------- | -------- | -------- | --------- |
> | E5V                     | -          | Qwen2.5-VL 3B | 47.1          | 71.4          | 79.8           | 24.2     | 47.3     | 58.4      |
> | E5V                     | -          | Qwen2.5-VL 7B | 50.2          | 75.4          | 82.3           | 27.0     | 50.8     | 61.6      |
> | VLM2Vec                 | 662K       | Qwen2-VL 7B   | 69.3          | 90.3          | 94.3           | 47.2     | 72.6     | 81.1      |
> | FreeRet-embed           | -          | Qwen2.5-VL 3B | 55.0          | 79.4          | 86.0           | 29.8     | 54.4     | 65.4      |
> | FreeRet (Rerank top 20) | -          | Qwen2.5-VL 3B | 77.0          | 89.6          | 90.8           | 49.8     | 69.5     | 74.0      |
> | FreeRet (Rerank top 50) | -          | Qwen2.5-VL 3B | 79.1          | 93.2          | 94.9           | 52.0     | 75.2     | 81.8      |
>
> | Text Retrieval          | Train data | MLLM          | Flickr30K R@1 | Flickr30K R@5 | Flickr30K R@10 | COCO R@1 | COCO R@5 | COCO R@10 |
> | ----------------------- | ---------- | ------------- | ------------- | ------------- | -------------- | -------- | -------- | --------- |
> | E5V                     | -          | Qwen2.5-VL 3B | 59.3          | 81.3          | 88.4           | 35.6     | 59.7     | 70.5      |
> | E5V                     | -          | Qwen2.5-VL 7B | 60.3          | 82.9          | 89.9           | 35.7     | 60.6     | 71.6      |
> | VLM2Vec                 | 662K       | Qwen2-VL 7B   | 87.4          | 97.7          | 99.1           | 60.6     | 82.7     | 89.6      |
> | FreeRet-embed           | -          | Qwen2.5-VL 3B | 67.4          | 88.8          | 93.2           | 42.0     | 67.4     | 77.2      |
> | FreeRet (Rerank top 20) | -          | Qwen2.5-VL 3B | 79.9          | 94.3          | 95.7           | 55.3     | 79.6     | 84.1      |
> | FreeRet (Rerank top 50) | -          | Qwen2.5-VL 3B | 80.8          | 96.8          | 97.9           | 53.8     | 81.3     | 88.0      |
>
> > **Q5: Although controlled prompts suppress function words, the differences in common word frequencies among different languages are relatively large. Have the authors considered generating within a restricted task vocabulary set? If run on Chinese or mixed corpora, what impact does lemmatization have on performance?**
>
> Thank you for the insightful comments.
>
> We agree that constructing a restricted task vocabulary (e.g., topic lists / category sets) can effectively suppress function words and reduce cross‑lingual drift, especially in domain‑specific settings. Such predefined concept sets can also inject task‑specific priors that help guide the model’s attention. However, this strategy becomes less practical in general‑purpose retrieval: (1) a sufficiently broad concept set must be very large, substantially increasing input length and computation, and (2) any fixed set risks missing concepts required by unanticipated conditions. In contrast, our direct‑instruction prompting remains flexible and task‑agnostic, making it better suited for open‑domain usage, whereas predefined sets are more appropriate when the downstream task is known.
>
> Regarding cross‑lingual scenarios: we mitigate cross‑lingual word distribution drift by providing all instructions in English, which effectively anchors the model to generate English concepts even when the underlying data are in other languages. As noted in our response to **Reviewer 5R9b (Q2)**, we evaluated FreeRet on our translated MMEB‑Chinese and MMEB‑French while keeping the English instructions unchanged. The model maintained stable performance across languages, indicating that language variation do not materially affect our method under this setup.

---

> ### Author Response · Authors · 2025-11-23
>
> > **Q6: The option text and explanatory prompt of MCQ may affect the coupling of the model's "reasoning - judgment". Have the authors systematically compared the effects of different MCQ copywriting (more neutral vs. more fine grain)? Is there an inflated result caused by "overly strong prompts"?**
>
> Thank you for your insightful comments.
>
> In our original experiments, the option text was either concise choices (e.g., “A. Yes B. No”) or more descriptive choices (e.g., “A. The caption matches the image content. B. The caption does not match the image content”), which were manually designed and randomly assigned. We agree that a clearer distinction between option types is important.
>
> To assess the impact of MCQ copywriting, we conducted an additional controlled study with two standardized variants: (i) Neutral: strictly “A. Yes B. No”; (ii) Fine‑grained: GPT‑5 rewrites both positive and negative choices into single‑sentence descriptions. We evaluated both variants on MMEB with a reranking pool size of 10. Results are summarized below.
>
> | MCQ copywriting        | Classification | VQA   | Retrieval | Grounding | Average |
> | ---------------------- | -------------- | ----- | --------- | --------- | ------- |
> | Used in original paper | 67.18          | 67.62 | 66.31     | 74.28     | 67.80   |
> | Neutral option         | 67.01          | 70.56 | 65.43     | 71.25     | 67.94   |
> | Fine-grained option    | 65.31          | 71.42 | 65.53     | 74.15     | 68.06   |
>
> We observe that neutral options slightly favor classification, whereas fine‑grained options improve grounding. Importantly, the manually written options used in our submission underperform both standardized variants in VQA and in the overall average, indicating that our reported results are not inflated by overly strong prompts.
>
> We will include these analyses and clarifications in the revised paper.
>
> > **Q7: It is mentioned in [1] that the responses of LLMs to MCQ are actually unstable. How do the authors view this issue?**
>
> Thank you for pointing out this relevant work.
>
> [1] provide an insightful analysis showing that earlier-generation LLMs (e.g., LLaMA, Vicuna, Falcon, GPT‑3.5) exhibit option‑ID selection bias when answering MCQs. Our findings concern a *different empirical regime*: we observe that the Qwen2.5‑VL family demonstrates stable option selection under context-free instructions.
>
> Why do the observations differ? We believe this discrepancy is due to the substantial evolution of training practices. The models examined in [1] primarily belong to the GPT‑3.5 era, where the LLM training **emphasized broad world knowledge** rather than problem solving or instruction-following. MCQ-style formats were relatively scarce in their training data, leading to instability.
>
> In contrast, modern MLLMs such as Qwen2.5‑VL (contemporaneous with GPT‑4.5 / OpenAI o1) undergo significantly richer post-training on complex reasoning, structured evaluation tasks, and user-facing instruction data, such formats include abundant MCQ-style supervision. This increased exposure appears to make contemporary models substantially more robust and less prone to option‑ID bias, consistent with our empirical results.
>
> ---
>
> [1] Large language models are not robust multiple choice selectors. ICLR 2024.

---

> > ### Comment · Reviewer_e7fh · 2025-11-25
> >
> > Thank you for your detailed rebuttal. I've decided to maintain my rating.

---

> > > ### Author Response · Authors · 2025-11-25
> > >
> > > We appreciate your recognition and insightful comments. Please let us know if you have any further questions.

---

### Official Review · Reviewer_8Y5y · 2025-10-31

**Soundness:** 1
**Presentation:** 3
**Contribution:** 2
**Rating:** 2
**Confidence:** 5

**Summary:**

This work aims to convert MLLM into retrievers by training-free methods, where the authors claim that fine-tuning MLLM needs too-much data source and computation, and would show little generalization. Their proposed FreeRet provide three advcances: 1) refine embedding quality by bypassing the final MLP before LM head of MLLM, 2) stabilize the embedding space via controlled summarization prompts, and 3) cast reranking as multiple-choice questions to mitigate LLM framing effect. By combining the FreeRet embedding and reranking models,  FreeRet surpass a range of heavly fine-tuned embedding models on MMEB, which is a unfair comparson. I like the paper many points, such as probing experiments on lexicalization pressure. But I think the experiment part leads to a unscientific framework and findings.

**Strengths:**

1. The idea of training-free multimodal retriever approach, which is very practical in many low-resource domains and scenarios.
2. The probing of lexicalization pressure in MLLM representation, very well thinking.

**Weaknesses:**

1. The model performance contribution is mixed, FreeRet is a retreival-stack of embedding+reranking, comparing to solely embedding models is clearly unfair. I think is absolutely a big issue.
2. The research contribution is somehow unclear, we can not know the effectiveness of embedding part of FreeRet.
3. The reranking part (LLM framing effect) is not a scientific or technical contribution IMHO, just a prompt engneering.

**Questions:**

1. I would suggest authors to show the embedding result on main table 1 & 2.
2. It's would be helpful to understand the contribution by comparing with some zero-shot rerank baselines, such as ones in MM-Embed [1].

[1] MM-Embed: Universal Multimodal Retrieval with Multimodal LLMs (Lin et al, ICLR 2025)

---

> ### Author Response · Authors · 2025-11-23
>
> > **Q1: The model performance contribution is mixed, FreeRet is a retreival-stack of embedding+reranking, comparing to solely embedding models is clearly unfair.**
>
> Thank you for raising this point.
>
> We would like to clarify that FreeRet does not stack **two separate models**. Instead, it **unifies embedding and reranking within a single MLLM**, using the same forward function and the same model parameters. In contrast, the compared embedding-only models **cannot perform reranking at all** without an additional reranker, meaning they inherently require **two separate models** to achieve the same functionality. From this perspective, the comparison is fair: **each method uses only one model at inference time**.
>
> Additionally, FreeRet is **entirely training‑free**, while other methods rely on **millions of curated multimodal pairs** and task-specific supervision. This puts FreeRet at a disadvantage in terms of optimization and data exposure, making its strong performance even more noteworthy.
>
> To further address your concern regarding matching the embedding+reranking setting, we provide additional experiments under fully aligned conditions. We reproduced MM‑Embed [1] and LamRA [2], two embedding plus reranking methods, using their official checkpoints and carefully crafted prompts following their papers and codebases. We also include UniME‑V2 [3], a state-of-the-art retriever directly optimized on MMEB. As shown below, even without any training and while unifying embedding and reranking within one model, **FreeRet substantially outperforms models trained on millions of samples**, and is only slightly behind UniME-V2, which is directly optimized on MMEB with advanced training skills.
>
> We hope the clarification and additional evidence alleviate your concerns.
>
> | Method   | Foundation (embed+rerank)     | Train Data                                | Reranking       | Precision@1 on MMEB |
> | -------- | ----------------------------- | ----------------------------------------- | --------------- | ------------------- |
> | MM-Embed | NV-Embed 7B + LLaVA-Next 7B   | M-BEIR (1.1M) + 11 T2T retrieval datasets | Pointwise Top10 | 54.9                |
> | LamRA    | Qwen2.5-VL 7B + Qwen2.5-VL 7B | NLI (273k) + M-BEIR (1.1M)                | Pointwise Top10 | 55.0                |
> | UniME-V2 | Qwen2-VL 7B + Qwen2.5-VL 7B   | MMEB (662k)                               | Listwise Top5   | 69.6                |
> | FreeRet  | Qwen2.5-VL 7B                 | -                                         | Pointwise Top10 | 67.8                |
>
> > **Q2: The research contribution is somehow unclear, we can not know the effectiveness of embedding part of FreeRet. I would suggest authors to show the embedding result on main table 1 & 2.**
>
> Thank you for your comment.
>
> The effectiveness of the embedding component of FreeRet is already demonstrated in **Fig. 5**: the *w/o reranking* setting corresponds exactly to using the embedding stage alone. Its performance remains around 50–54, closely approaching GME, the SOTA zero‑shot embedder trained on 8M multimodal pairs. We originally omitted these numbers from Tab. 1 to avoid clutter, but we have now added an explicit reference in the Tab. 1 caption to make this connection clearer. Please refer to the revised paper.
>
> Following your suggestion, we have also added the embedding‑only results (FreeRet‑embed) to Tab. 2 of the revised paper. As shown, FreeRet‑embed still outperforms strong specialist embedding baselines by a noticeable margin, highlighting the intrinsic strength and generalizability of FreeRet even without reranking.
>
> ---
>
> [1] MM-Embed: Universal Multimodal Retrieval with Multimodal LLMs. ICLR 2025.
>
> [2] Lamra: Large multimodal model as your advanced retrieval assistant. CVPR 2025.
>
> [3] UniME-V2: MLLM-as-a-Judge for Universal Multimodal Embedding Learning. AAAI 2026.

---

> ### Author Response · Authors · 2025-11-23
>
> > **Q3: The reranking part (LLM framing effect) is not a scientific or technical contribution, just a prompt engneering.**
>
> Thank you for the comment.
>
> We would like to clarify that our contribution extends beyond proposing a different prompt. Our central finding is the **identification and analysis of a systematic framing effect** in LLM-based reranking, an effect that, to our knowledge, has not been documented in prior multimodal retrieval literature despite the widespread use of LLM rerankers.
>
> Our experiments show that logically equivalent binary frames can induce up to 5% accuracy variance, revealing **model‑internal sensitivities rather than superficial prompt‑engineering artifacts**. To probe the source, we conduct context‑free bias tests demonstrating that certain label tokens carry asymmetric prior logits. These biases can reliably reflect downstream performance, indicating a real pretraining‑induced imbalance rather than noise from prompt design.
>
> This matters scientifically because such brittleness directly affects the **reliability and reproducibility of LLM rerankers**, a critical component of modern retrieval systems. Our MCQ-style formulation is therefore not ad‑hoc prompt tuning but a principled mitigation strategy: it neutralizes lexical affect and aligns the task with formats the models commonly encounter during pretraining.
>
> We will clarify this distinction in the revision.
>
> > **Q4: It's would be helpful to understand the contribution by comparing with some zero-shot rerank baselines, such as ones in MM-Embed [1].**
>
> Thank you for your constructive suggestions.
>
> Following your recommendation,  we conducted a comprehensive comparison between FreeRet and existing *embedding + reranking* baselines. In particular, we included MM‑Embed [1], LamRA [2], and UniME‑V2 [3].
>
> As detailed in our response to your **Q1**, FreeRet consistently outperforms MM‑Embed and LamRA on MMEB, even though they rely on millions of data pairs for fine‑tuning. FreeRet also achieves performance comparable to methods that are directly optimized on the benchmark.
>
> Additionally, as noted in our response to **Q2 of Reviewer 5R9b**, we constructed a **multilingual MMEB** variant using Qwen3‑14B (covering Chinese and French). FreeRet maintains consistent advantages over LamRA across all language settings.
>
> Finally, regarding computational efficiency (response to **Q3 of Reviewer 5R9b**), FreeRet benefits from unifying embedding and reranking within a single model. This allows partial sharing of the forward pass, resulting in clear lower inference cost compared to separate embedding‐and‐reranker pipelines.
>
> We hope these additional comparisons clarify FreeRet’s contributions relative to other reranking baselines.
>
> ---
>
> [1] MM-Embed: Universal Multimodal Retrieval with Multimodal LLMs. ICLR 2025.
>
> [2] Lamra: Large multimodal model as your advanced retrieval assistant. CVPR 2025.
>
> [3] UniME-V2: MLLM-as-a-Judge for Universal Multimodal Embedding Learning. AAAI 2026.

---

> ### Comment · Reviewer_8Y5y · 2025-11-26
>
> ### Re: Response for Q1
> Thanks for the detailed clarification. However, I don’t think the argument that “each method uses only one model at inference time” is actually meaningful here. Embedding-based retrieval and point-wise reranking are fundamentally different techniques.  The former scales as O(Q + D), while the latter scales as O(Q × D). Reranking enjoys substantial modeling and computational advantages that embedding-only methods simply cannot match.
>
> I do appreciate the contribution of unifying embedding and reranking into a single model, that’s interesting. But avoiding the fact that reranking has inherent advantages, and then comparing FreeRet only against embedding-based models, ends up being misleading for readers trying to understand the true contribution.
>
> ---
>
> ### Re: Response for Q2
> Thanks for adding FreeRet-embed to Table 2. It’s helpful. I’m wondering why the same update wasn’t applied to Table 1. Table 1 already marks E5-V as training-free, which makes it the ideal baseline system for comparing to FreeRet-embed. Showing FreeRet-embed there (with a clear caption that it’s the embedding-only setting) would make the comparison straightforward and transparent. As it stands, leaving FreeRet-embed out of Table 1 while presenting other training-free baselines can give readers an incomplete impression of the method’s component-level performance. Could you please add FreeRet-embed to Table 1 as well?
>
> ---
>
> ### Re: Response for Q3&4
> Thanks for the additional experiments. They are very helpful.
>
> ---
>
> Overall, I like the idea behind this work. But the way the experiments are currently presented gives me the impression that the comparisons are intentionally tilted. I hope the revision can make the experimental comparisons more transparent and explicit, so that readers can properly appreciate the strengths of your approach.

---

> ### Author Response · Authors · 2025-11-26
>
> Thank you for your detailed response and for recognizing our contributions.
>
> We agree that the reranking stage is much more computationally intensive than the embedding stage, and methods with different retrieval pipelines should be distinguished accordingly.
>
> Following your suggestion, we have updated Table 1 by:
> 1. Adding **FreeRet‑embed** to compare against embedding‑only methods;
> 2. Adding **three embedding–reranking baselines** for a fair comparison with FreeRet;
> 3. Clearly separating the two retrieval pipelines into different table regions.
>
> From the revised Table 1, we observe that **FreeRet‑embed** achieves performance comparable to models trained on millions of paired examples. Relative to our baseline **E5‑V**, FreeRet‑embed improves by **13.9%**. With reranking, **FreeRet** achieves further substantial gains, outperforming existing recall‑rerank baselines by a large margin (**12.8%**) and reaching performance comparable to UniME‑V2, which is directly optimized on MMEB.
>
> Beyond strong retrieval quality, FreeRet offers several training‑free advantages (Sec. 4.3): instant deployment, omni‑modality retrieval, preservation of autoregressive capabilities, unified model for embedding&reranking, and applicability to RAG and long‑video understanding. It is also more computationally friendly than other embedding‑reranking methods (response to Q3 of Reviewer 5R9b), generalizes well across model families and scales (Table 6), maintains strong video performance (Table 2), and handles cross‑lingual scenarios effectively (response to Q1 of Reviewer 5R9b). We hope these clarifications help contextualize the contributions of FreeRet.
>
> Thank you again for your time and constructive feedback. We are happy to provide any further clarifications.

---

> > ### Comment · Reviewer_8Y5y · 2025-11-27
> >
> > Thank you for the response. I suggest clearly present and dicsuss the embedding and reranking separately, and show the efficiency analysis in revision.
> > I raise my rating.

---

> ### Author Response · Authors · 2025-11-27
>
> Thank you for your response. We sincerely appreciate your positive update.
>
> Following your suggestions, we have
> 1. Significantly revised Section 4.2 to provide a clear separation between embedding and reranking discussions;
> 2. Added Appendix A.8 to present the efficiency analysis, including comparisons with baseline methods and an ablation study on compute resources across different model scales and reranking candidate sizes.
>
> These changes are included in the latest revision. We are happy to address any further suggestions or questions.

---

### Official Review · Reviewer_5R9b · 2025-10-31

**Soundness:** 3
**Presentation:** 3
**Contribution:** 3
**Rating:** 6
**Confidence:** 3

**Summary:**

The paper proposes a plug-and-play framework that turns any pretrained multimodal large language model (MLLM) into a two-stage retriever without further training.

**Strengths:**

- It introduces a fully parameter-free retrieval framework that repurposes any existing MLLM as an effective retriever, avoiding the cost of multimodal fine-tuning.
- The work identifies bottlenecks such as lexicalization pressure in the final MLP, and reveals the LLM framing effect in reranking.
- Unlike fine-tuned retrievers, it retains the original model’s multimodal understanding and generative capabilities.

**Weaknesses:**

- Limited novelty, although the training-free idea is appealing, the improvements mostly in prompt design.
- The method’s success relies heavily on manually crafted prompts and framing choices, which may not generalize well across unseen domains or languages without careful tuning.
- Running the same large MLLM for both embedding and reranking can be computationally expensive at inference time, yet the paper provides little discussion on latency or resource cost.
- The insights provided by the Probing Experiment are limited. I believe it is expected that the final layer undergoes a transformation of the representation space, so the experiment offers only weak support for the claim of lexicalization pressure. Moreover, in the analysis of Semantic Retention, using only synonym pairs is insufficient to demonstrate that semantic resolution degrades after the LM head. Including distance analyses among synonym, antonym, and random pairs would make the argument more convincing. In addition, I think a deeper analysis and explanation are needed to clarify why lexicalization pressure leads to performance degradation.

**Questions:**

I suggest that the authors further analyze and investigate the generalization ability of the proposed prompts, in order to verify their robustness and effectiveness in open-ended or noisy retrieval scenarios. In addition, the relationship between lexicalization pressure and model performance requires further investigation.

---

> ### Author Response · Authors · 2025-11-23
>
> > **Q1: Limited novelty, although the training-free idea is appealing, the improvements mostly in prompt design.**
>
> Thank you for your comment.
>
> While our approach is training-free, it goes beyond simple prompt design. The core contribution of our study is to **systematically re-examine the intrinsic retrieval capabilities of MLLMs,** challenging the prevailing assumption that strong retrieval performance necessarily requires task-specific data curation and model fine-tuning.
>
> Prior studies (e.g., E5-V) generally show that naive prompting yields sub-optimal retrieval accuracy, motivating researchers to retrain models. In contrast, our work offers a principled and interpretable framework for direct, zero-shot retrieval with MLLMs. To the best of our knowledge, **we are the first to demonstrate that existing MLLMs, when guided appropriately, already possess substantial retrieval competence**.
>
> We hope our study can inspire further exploration of the inherent capability of pretrained MLLMs rather than immediately resorting to additional fine-tuning.
>
> > **Q2: The method’s success relies heavily on manually crafted prompts and framing choices, which may not generalize well across unseen domains or languages without careful tuning.**
>
> Thank you for your comment.
>
> First, we would like to clarify that FreeRet requires only **lightweight** prompt design. The **sole customization** needed is a short description of the retrieval goal, which is inherent to any retrieval system. For instance, in N24News, the embedding-stage task-alignment prompt is simply: “You are required to assess if the news is related to the topic.” Prompts for semantic grounding and noise suppression remain similar across tasks. The reranking prompt extends the same goal description with two options (A/B) and does not involve additional tuning beyond defining the task itself. Importantly, specifying the retrieval objective is **fundamental and unavoidable** for precise retrieval, and existing systems adopt similar designs (e.g., MM-Embed [1] Tab. 16–18; LamRA [2] Tab. 12).
>
> Second, to evaluate the cross-lingual generalization you are concerned about, we translated the MMEB benchmark into *Chinese* and *French* using Qwen3-14B. We then compared FreeRet with LamRA, which fine-tunes Qwen2.5-VL on 1.4M high‑quality multimodal pairs for both embedding and reranking. Both methods **kept exactly the same prompts as in the English setting**. As shown below, FreeRet demonstrates strong cross‑lingual robustness without any prompt modification. The moderate performance drop is expected and likely arises from translation imperfections, inconsistencies between translated text and the English text in the images, and Qwen2.5‑VL’s uneven multilingual proficiency.
>
> |                                 | MLLM          | Train data | MMEB (English) | MMEB (Chinese) | MMEB (French) |
> | ------------------------------- | ------------- | ---------- | -------------- | -------------- | ------------- |
> | LamRA (Embed + Rerank Top 10)   | Qwen2.5-VL 7B | 1.4M       | 55.0           | 48.3           | 45.7          |
> | FreeRet (Embed + Rerank Top 10) | Qwen2.5-VL 7B | -          | 67.8           | 62.7           | 60.4          |
>
> ---
>
> [1] MM-Embed: Universal Multimodal Retrieval with Multimodal LLMs. ICLR 2025.
>
> [2] Lamra: Large multimodal model as your advanced retrieval assistant. CVPR 2025.

---

> ### Author Response · Authors · 2025-11-23
>
> > **Q3: Running the same large MLLM for both embedding and reranking can be computationally expensive at inference time, yet the paper provides little discussion on latency or resource cost.**
>
> Thank you for raising this point.
>
> We have added a detailed analysis of latency and resource usage. Averaged over all MMEB test samples with batch size set to 1, we measure peak GPU memory, per‑sample latency for FreeRet's embedding stage and full‑pipeline (embedding then reranking the top‑10 candidates). We compare FreeRet with two strong baselines, MM‑Embed [1] and LamRA [2], both of which require separate models for embedding and reranking.
>
> Our results show that FreeRet achieves superior retrieval performance while being substantially more efficient. Because prior methods rely on two distinct models, their memory consumption is almost doubled. In contrast, FreeRet unifies embedding and reranking within a single MLLM. This design allows the reranking stage to reuse both the vision‑encoder outputs and the shared‑prefix KV‑cache, significantly reducing computational overhead.
>
> | Method       | Foundation (embed+rerank)   | Unify | Train data | GPU Memory (embed+rerank) | Latency (embed) | Precision@1 (embed) | Latency (embed+rerank) | Precision@1 (embed+rerank) |
> | ------------ | --------------------------- | ----- | :--------: | ------------------------- | --------------- | ------------------- | ---------------------- | -------------------------- |
> | MM-Embed [1] | NV-Embed7B + LLaVA-Next 7B  | ✗     |    1.1M    | 33.28 GB                  | 0.11 s          | 52.8                | 1.51 s                 | 54.9                       |
> | LamRA [2]    | Qwen2.5VL 7B + Qwen2.5VL 7B | ✗     |    1.4M    | 30.91 GB                  | 0.08 s          | 51.3                | 0.78 s                 | 55.0                       |
> | FreeRet      | Qwen2.5VL 3B                | ✓     |     -      | 9.01 GB                   | 0.08 s          | 50.9                | 0.35 s                 | 59.9                       |
> | FreeRet      | Qwen2.5VL 7B                | ✓     |     -      | 17.75 GB                  | 0.08 s          | 53.7                | 0.47 s                 | 67.8                       |
>
> > **Q4: I believe it is expected that the final layer undergoes a transformation of the representation space, so the experiment offers only weak support for the claim of lexicalization pressure.**
>
> Thank you for your comment.
>
> We agree that a final transformation of the representation space is expected. However, our contribution goes beyond simply confirming its existence: we aim to characterize **when** this transformation occurs and **where** it guides the representations. To this end, we evaluate per-layer hidden‑state similarity (blue curve in Fig. 2a) to track representation changes, and we simultaneously measure the **LM head alignment degree** (orange curve in Fig. 2a), quantified as the cosine similarity between hidden states and LM-head weights .
>
> Both measures remain stable across early and middle layers and shift sharply only at the final layer. This indicates that the transformation is confined to the last MLP layer, at which point the representations become explicitly aligned with the lexical space defined by the LM head. We interpret this emergent alignment as **quantitative evidence of lexicalization pressure**. Moreover, this pattern is statistically robust: it is averaged over 720 samples across 36 MMEB datasets and is consistently observed across three model scales (3B, 7B, 32B).
>
> ---
>
> [1] MM-Embed: Universal Multimodal Retrieval with Multimodal LLMs. ICLR 2025.
>
> [2] Lamra: Large multimodal model as your advanced retrieval assistant. CVPR 2025.

---

> ### Author Response · Authors · 2025-11-23
>
> > **Q5: Using only synonym pairs is insufficient to demonstrate that semantic resolution degrades after the LM head. Including distance analyses among synonym, antonym, and random pairs would make the argument more convincing.**
>
> Thank you for the insightful suggestion.
>
> Following your recommendation, we conducted an extended analysis comparing hidden‑state similarities among synonym, antonym, and random word pairs. **The new results are shown in Fig. 8**, and the corresponding discussion **has been added to Sec. A.1** in the revised paper.
>
> The additional evaluation shows: before the final MLP layer, representations maintain clear semantic structure, with synonym pairs most similar, antonym pairs moderately similar, and random pairs least similar. However, this structure collapses immediately after the final MLP layer, where the similarities of all three categories converge. This provides additional evidence that lexicalization pressure in the final MLP layer degrades semantic fidelity, consistent with the performance decline observed in our ablations.
>
> We appreciate your feedback, which helped us substantially strengthen the empirical support for our conclusions.
>
> > **Q6: A deeper analysis and explanation are needed to clarify why lexicalization pressure leads to performance degradation.**
>
> Thank you for raising this point. We expand our analysis below.
>
> In our work, lexicalization pressure refers to the transformation that adapts hidden states for discrete token prediction. Our probing results (Fig. 2a) reveal two convergent patterns: (i) representational continuity across layers collapses sharply at the last MLP, and (ii) alignment with the lexical head rises abruptly at the same point. These jointly indicate that the final MLP explicitly **restructures semantically rich features into a geometry optimized for vocabulary logits**, prioritizing directions that aligned with specific lexical choices and consequently suppressing semantic components that are not directly predictive of the next token.
>
> Our analyses of synonym‑pair similarity (Fig. 2b) further show a sharp reduction in similarity among semantically related concepts; while random-pair similarity also declines (Fig. 8), the drop for synonyms is substantially larger. Moreover, the similarity patterns among synonym, antonym, and random pairs (Fig. 8) demonstrate a clear loss of semantic separation. **These findings jointly illustrate how lexicalization pressure erodes semantic retention**, which is crucial for retrieval.
>
> Empirically, **removing only the final MLP layer yields substantial improvements** on the MMEB benchmark (Tab. 3a), with 5.33% and 5.71% average accuracy gains across 36 datasets. This provides direct evidence that lexicalization pressure contributes to downstream performance degradation.
>
> Finally, we provide a theoretical justification in Sec. A.2: the gradient of the LM loss lies entirely within the span of the LM-head vectors, implying that optimization controls only the projection of the hidden state onto this lexical subspace. The final MLP therefore learns to map its input into this subspace, amplifying alignment with the predicted token while diminishing components that encode broader semantics. Discarding the final MLP bypasses this forced projection.
>
> We hope these clarifications adequately address your concerns.

---

> > ### Comment · Reviewer_5R9b · 2025-11-25
> >
> > I appreciate the authors’ response. It addresses the major concerns, so I am maintaining my positive rating.

---

> > > ### Author Response · Authors · 2025-11-25
> > >
> > > Thank you for your detailed and insightful reviews, and for recognizing the contributions of our paper. Should you have any further concerns or require additional clarification, please do not hesitate to ask.

---

### Meta-Review · Area_Chair_DAug · 2025-12-26

**Summary:**

This work provides an empirical analysis of prompting strategies that can be used to perform semantic search using pre-trained off-the-shelf language models, without additional training. Namely, different prompting approaches are compare to either embed input content as well as to carry out re-ranking, in which case authors discuss different ways of posing the re-ranking problem, and show that choice to significantly affect performance downstream. Evaluation on standard benchmarks show surprisingly strong performance if one conditions on the fact that no training is carried out, although controlling for inference cost could be perhaps a concern as will be further detailed.

**Reviewer Concerns:**

The main concerns brought up by the reviewers were related to limited novelty, as the paper claims to be introducing a new method, but offers a prompting strategy. Perhaps those concerns would disappear had the paper been framed as an analysis paper, showing the effect of different prompting strategies in search performance. Besides that, reviewers mentioned issues regarding the presentation, as the text reads a bit high level and many implementation details are missing. For instance, the text doesn't clearly detail how embeddings are computed other than mentioning tweaks in the prompt the authors observed to work well and layers where embeddings are collected. I would add that dropping the last layer is somewhat common practice when training encoders in the visual domain for instance, and the effect the authors dedicated a large chunk of the paper to describe, i.e., the shift between the language model training task and the task the representations will be used for, is somewhat expected. The evaluation is also a bit unclear when it comes to assessing inference costs. Authors did add results during the rebuttal, but I would say the scenario is still a bit unclear and would require careful discussion in the paper. For instance, typical search applications pre-encode targets and store them into a vector database, and re-ranking costs are somewhat the same for proposed models and baselines, so the differences presented during the rebuttal require a bit more discussion and clarification as do many other parts of the paper (e.g., LamRA and FreeRet at 7B should have similar latency). Finally, concerns were raised regarding the extent to which the proposed prompting generalises to new datasets and models, which was only partially addressed during the discussion.

**Reviewer Scores:**

2 out of 3 reviewers had positive leaning scores, but raised important concerns that I don't think were fully addressed during the rebuttal, and would require a major refactor of the paper in order for them to be addressed (lack of novelty, clarity issues, lack of performance comparisons). I would say those reviewers would maintain their original scores, which was positive.

The more negative leaning reviewer shared some of the concerns raised above, and additonaly wasn't convinced by the argument and empirical evidence presented for the lexical matching pressure due to the underlying pre-training task. The reviewer was somewhat satisfied by the authors's response and mentioned they would raiser their score, though the final recommendation is a bit unclear.

I personally tend to side with the more negative leaning results as the results in the paper seem a bit preliminary and require expansion, especially along the performance axis, and the extent to which the proposal generalizes. The presentation also needs significant improvements prior to publication in my opinion.

---

### Decision · Program_Chairs · 2026-01-26

Reject